# Space and Time Efficient Kernel Density Estimation in High Dimensions

**Arturs Backurs**[*]
TTIC
backurs@ttic.edu

**Piotr Indyk**
MIT
indyk@mit.edu

**Tal Wagner**
MIT
talw@mit.edu

## Abstract

Recently, Charikar and Siminelakis (2017) presented a framework for kernel density estimation in provably sublinear query time, for kernels that possess a certain hashing-based property. However, their data structure requires a significantly increased super-linear storage space, as well as super-linear preprocessing time. These limitations inhibit the practical applicability of their approach on large datasets.

In this work, we present an improvement to their framework that retains the same query time, while requiring only linear space and linear preprocessing time. We instantiate our framework with the Laplacian and Exponential kernels, two popular kernels which possess the aforementioned property. Our experiments on various datasets verify that our approach attains accuracy and query time similar to Charikar and Siminelakis (2017), with significantly improved space and preprocessing time.

## 1 Introduction

Kernel density estimation is a fundamental problem with many applications in statistics, machine learning and scientific computing. For a *kernel function* $k : \mathbb{R}^d \times \mathbb{R}^d \to [0,1]$, and a set of points $X \subset \mathbb{R}^d$, the kernel density function of $X$ at a point $y \in \mathbb{R}^d$ is defined as:[2]

$$\text{KDE}_X(y) = \frac{1}{|X|} \sum_{x \in X} k(x,y).$$

Typically the density function is evaluated on a multiple queries $y$ from an input set $Y$. Unfortunately, a naïve exact algorithm for this problem runs in a rectangular $O(|X||Y|)$ time, which makes it inefficient for large datasets $X$ and $Y$. Because of this, most of the practical algorithms for this problem report approximate answers. Tree-based techniques [GS91, GM01, GB17] lead to highly efficient approximate algorithms in low-dimensional spaces, but their running times are exponential in $d$. In high-dimensional spaces, until recently, the best approximation/runtime tradeoff was provided by simple uniform random sampling. Specifically, for parameters $\tau, \epsilon \in (0,1)$, it can be seen that if $X'$ is a random sample of $O\left(\frac{1}{\tau} \frac{1}{\epsilon^2}\right)$ points from $X$, then $\text{KDE}_{X'}(y) = (1 \pm \epsilon) \text{KDE}_X(y)$ with constant probability[3] as long as $\text{KDE}_X(y) \geq \tau$.

This approximation/runtime tradeoff was recently improved in [CS17], who proposed a framework based on *Hashing-Based Estimators (HBE)*. The framework utilizes locality-sensitive hash (LSH)

---

[*]Authors ordered alphabetically.

[2]We note that all algorithms discussed in this paper easily extend to the case where each term $k(x,y)$ is multiplied by a positive weight $w_x \geq 0$, see e.g., [CS17].

[3]The probability of correct estimation can be reduced to $1 - \delta$ for any $\delta > 0$ at the cost of increasing the sample size by a factor of $\log(1/\delta)$. Since the same observation applies to all algorithms considered in this paper, we will ignore the dependence on $\delta$ from now on.

Table 1: Comparison of runtime and space bounds. Notation: $\tau \in (0,1)$ denotes a lower bound for KDE values; $d$ denotes the dimension; $\epsilon \in (0,1)$ denotes the approximation error.[4]

| Algorithm | Query Time | # Stored hashes |
|---|---|---|
| Random Sampling | $O(d/\tau \cdot 1/\epsilon^2)$ | $O(1/\tau \cdot 1/\epsilon^2)$ |
| HBE | $O(d/\sqrt{\tau} \cdot 1/\epsilon^2)$ | $O(1/\tau^{3/2} \cdot 1/\epsilon^4)$ |
| This paper | $O(d/\sqrt{\tau} \cdot 1/\epsilon^2)$ | $O(1/\tau \cdot 1/\epsilon^2)$ |

functions [IM98], i.e., randomly selected functions $h : \mathbb{R}^d \to U$ with the property that for any $x, y \in \mathbb{R}^d$, the collision probability $\Pr_h[h(x) = h(y)]$ is "roughly" related to the kernel value $k(x, y)$. HBE reduces the evaluation time to (about) $O\left(\frac{1}{\sqrt{\tau}} \frac{1}{\epsilon^2}\right)$. A recent empirical evaluation of this algorithm [SRB$^+$19] showed that it is competitive with other state of the art methods, while providing significant (up to one order of magnitude) runtime reduction in many scenarios.

One drawback of HBE approach, however, is its space usage, which is super-linear in the dataset size. Specifically, the algorithm constructs $O\left(\frac{1}{\sqrt{\tau}} \frac{1}{\epsilon^2}\right)$ hash tables, and stores the hash of each data point in each table. Consequently, the additional storage required for the hashes is proportional to the number of tables times the number of data points. As mentioned above, we can uniformly subsample the dataset down to $O\left(\frac{1}{\tau} \frac{1}{\epsilon^2}\right)$ points, leading to an overall space usage of $O\left(\frac{1}{\tau^{3/2}} \frac{1}{\epsilon^4}\right)$, which is $O\left(\frac{1}{\sqrt{\tau}} \frac{1}{\epsilon^2}\right)$ times that of the simple random sampling approach. The increase in storage also effects the preprocessing time of the HBE data structure, which requires $O\left(\frac{1}{\tau^{3/2}} \frac{1}{\epsilon^4}\right)$ hashes computations due to having to store every point in every table. As $\tau$ and $\epsilon$ can be very close to zero in practice, these drawbacks may pose a substantial bottleneck in dealing with large datasets.

**Our results.** In this paper we show that the super-linear amount of storage is in fact *not needed* to achieve the runtime bound guaranteed by the HBE algorithm. Specifically, we modify the HBE algorithm in a subtle but crucial way, and show that this modification reduces the storage to (roughly) $O\left(\frac{1}{\tau} \frac{1}{\epsilon^2}\right)$, i.e., the same as simple random sampling. Table 1 summarizes the performance of the respective algorithms. Our main result is the following theorem.

**Theorem 1.** *Let $k(x, y)$ be a kernel function, for which there exists a distribution $H$ of hash functions and $M \geq 1$ such that for every $x, y \in \mathbb{R}^d$,*

$$M^{-1} \cdot k(x,y)^{1/2} \leq \Pr_{h \sim H}[h(x) = h(y)] \leq M \cdot k(x,y)^{1/2}. \tag{1}$$

*There exists a data structure for Kernel Density Estimation with the following properties:*

- *Given a dataset $X \subset \mathbb{R}^d$ and parameters $\tau, \epsilon \in (0,1)$, we preprocess it in $O\left(\frac{1}{\tau} \cdot \frac{T_H M^3}{\epsilon^2}\right)$ time, where $T_H$ is the time to compute a hash value $h(x)$.*

- *The space usage of the data structure is $O\left(\frac{1}{\tau} \cdot \frac{(S_X + S_H)M^3}{\epsilon^2}\right)$, where $S_X$ is the space needed to store a point $x \in X$, and $S_H$ is the space needed to store a hash value $h(x)$.*

- *Given a query point $y$ such that $KDE_X(y) \geq \tau$, we can return with constant probability a $(1 \pm \epsilon)$-approximation of $KDE_X(y)$ in $O\left(\frac{1}{\sqrt{\tau}} \cdot \frac{(T_k + T_H)M^3}{\epsilon^2}\right)$ time, where $T_k$ is the time to compute a kernel value $k(x, y)$.*

We empirically evaluate our approach on the Laplacian kernel $k(x, y) = e^{-\|x-y\|_1/\sigma}$ and the exponential kernel $k(x, y) = e^{-\|x-y\|_2/\sigma}$. Both are commonly used kernels, and fit into the framework as

they satisfy the requirements of Theorem 1 with $M = O(1)$, $T_k = O(d)$, $T_H = O(\min\{d, d/\sigma\})$ and $S_H = O(\min\{d\log(1/\sigma), d/\sigma\})$, with high probability (over $h \sim H$). Our experiments confirm the analytic bounds and show that our approach attains a similar query time to approximation tradeoff as HBE, while using significantly less space and preprocessing time.

**Our techniques.** Our algorithm builds on the HBE approach of [CS17]. Recall that the algorithm selects $L = \Theta\left(\sqrt{1/\tau} \cdot 1/\epsilon^2\right)$ LSH functions $h_1 \ldots h_L$, and creates $L$ hash tables, such that for each $i = 1 \ldots L$, each point $x \in X$ is placed in the $j$th table in bin $h_j(x)$. To estimate $\mathrm{KDE}_X(y)$, the algorithm selects one point from each bin $h_1(y) \ldots h_L(y)$, and uses those points for estimation. To achieve the performance as in Table 1, the algorithm is applied to a random sample of size $s = O(1/\tau \cdot 1/\epsilon^2)$. The total space is therefore bounded by $O(sL) = O(1/\tau^{3/2} \cdot 1/\epsilon^4)$.

A natural approach to improving the space bound would be to run HBE on a smaller sample. Unfortunately, it is easy to observe that any algorithm must use at least $\Omega(1/\tau \cdot 1/\epsilon^2)$ samples to guarantee $(1 \pm \epsilon)$-approximation. Therefore, instead of sub-sampling the whole input to the HBE algorithm, we sub-sample the content of each hash table *independently* for each hash function $h_j, i = 1 \ldots L$. Specifically, for each hash function $h_j$, we include a point $x \in X$ in the $j$th hash table with probability $1/(s\sqrt{\tau})$. This reduces the expected number of stored hashes to $O(\sqrt{\tau}L)$. If we start from a sample of size $s = \Theta(1/\tau \cdot 1/\epsilon^2)$, then $\sqrt{\tau}L = O(s)$, yielding the desired space bound; at the same time, each point is included in at least one hash table with constant probability, which means that at least $\Omega(1/\tau \cdot 1/\epsilon^2)$ points will be included in the union of the hash tables with high probability. Perhaps surprisingly, we show that this increases the variance of the overall estimator by only a constant factor.

For an intuition of why subsampling by a factor $\sqrt{\tau}$ does not distort the kernel values by much, consider a simple setting where $\epsilon$ is a constant, $n = 1/\tau$, and there is only one data point $x$ that is very close to the query $y$ (contributing $\approx 1$) while all other points are far from $y$ (contributing $\approx 0$). In this case, the original HBE algorithm would collect the point $x$ from every bin $h_1(y) \ldots h_L(y)$, where $L = \sqrt{1/\tau}$. In contrast, if we subsample by a factor $\sqrt{\tau}$, then $x$ is expected to survive in one table, and thus our algorithm is still likely to identify one such bin in expectation. Conditioned on this event, the estimate of the algorithm is approximately correct. See more details in Section 3.

## 1.1 Related work

There is a vast amount of work on fast kernel density estimation in low dimensions, including the seminal Fast Gauss Transform [GS91] and other tree-based methods [GM01, GB17]. However, as mentioned above, they entail an exponential dependence on the input dimension. The tree-based ASKIT algorithm [MXB15] avoids this dependence and is suitable for the high-dimensional regime. However, it lacks rigorous guarantees on the approximation quality. The empirical evaluation in [SRB+19] showed that HBE is consistently competitive with ASKIT, and in some settings outperforms it by an order of magnitude.

Another important line of research has focused on *sparsifying* (reducing the size) of the input pointset while preserving kernel density function values. This can be accomplished by constructing *core-sets* [Phi13, ZJPL13, PT18] or related approaches [CWS12, SRB+19]. Although effective in low dimensions, in high dimensions such approaches require $\Omega(1/\epsilon^2)$ samples (for an additive error of $\epsilon > 0$ [PT18]), which is the same as the simple random sampling approach.[5] We note that the sparsification approach can be combined with our improvement, as we can run our algorithm on a core-set instead of the original data set, and retain the core-set size while speeding up the query time.

In addition to the aforementioned works of [CS17, SRB+19], LSH-based estimators have been applied in [CXS18, LS18b, WCN18, LS18a] to a variety of machine learning tasks.

## 2 Preliminaries

**Kernel Density Estimation.** Consider a kernel map $k : \mathbb{R}^d \times \mathbb{R}^d \to [0, 1]$. The kernel density estimation problem can be formally stated as follows.

**Definition 2.** *Let $X = \{x_1, \ldots, x_n\} \subset \mathbb{R}^d$ be an input dataset, and $\epsilon, \tau \in (0, 1)$ input parameters. Our goal is to construct a data structure such that for every query point $y \in \mathbb{R}^d$ that satisfies $KDE_X(y) \geq \tau$, we can return an estimate $\widetilde{KDE}(y)$ such that with constant probability,*

$$(1 - \epsilon)KDE_X(y) \leq \widetilde{KDE}(y) \leq (1 + \epsilon)KDE_X(y).$$

An exact computation of $KDE_X(y)$ performs $n$ kernel evaluations. By standard concentration inequalities, the above approximation can be achieved by evaluating the kernel $y$ with only $O(\frac{1}{\tau}\frac{1}{\epsilon^2})$ points chosen uniformly at random from $X$, and returning the average. As a result, we can assume without loss of generality (and up to scaling $\epsilon$ by a constant) that $n = O(\frac{1}{\tau}\frac{1}{\epsilon^2})$.

**LSHable kernels.** Locality-Sensitive Hashing (LSH) is a widely used framework for hashing metric datasets in a way that relates the collision probability of each pair of points to their geometric similarity. Kernel maps for which such hashing families exist are called "LSHable" [CK15]. The precise variant we will need is defined as follows.

**Definition 3.** *The kernel $k$ is called $(\frac{1}{2}, M)$-LSHable if there exists a family $H$ of hash functions $h : \mathbb{R}^d \to \{0, 1\}^*$, such that for every $x, y \in \mathbb{R}^d$, Equation (1) holds.*[6]

**Laplacian and Exponential kernels.** The Laplacian kernel is $k(x, y) = e^{-\|x-y\|_1/\sigma}$, where $\sigma > 0$ is the bandwidth parameter. The exponential kernel is defined similarly as $k(x, y) = e^{-\|x-y\|_2/\sigma}$ (the difference is in use of the $\ell_2$-norm instead of the $\ell_1$-norm). For our purposes the two are essentially equivalent, as they give the same analytic and empirical results. We will mostly focus on the Laplacian kernel, since as we will see, it is $(\frac{1}{2}, 1)$-LSHable. As a corollary, a random rotation of the dataset [DIIM04, CS17] can be used to show that the Exponential kernel is $(\frac{1}{2}, O(1))$-LSHable.

## 3 The Data Structure

We begin by recalling the HBE-based KDE data structure of [CS17]. For simplicity consider the case $M = 1$. During preprocessing, they sample $L = O(\frac{1}{\sqrt{\tau}\epsilon^2})$ hash functions $h_1, \ldots, h_L$ from the LSH family $H$, and store $h_j(x_i)$ for every $i = 1, \ldots, n$ and $j = 1, \ldots, L$. The preprocessing time is $O(T_H \cdot \frac{n}{\sqrt{\tau}\epsilon^2})$, and the space usage (in addition to the dataset) is $O(S_H \cdot \frac{n}{\sqrt{\tau}\epsilon^2})$, where $T_H$ is the time needed to evaluate the hash value of a point, and $S_H$ is the space needed to store it. Recalling we have assumed that $n = O(\frac{1}{\tau}\frac{1}{\epsilon^2})$, these become $O(T_H \cdot \frac{1}{\tau^{1.5}\epsilon^4})$ and $O(S_H \cdot \frac{1}{\tau^{1.5}\epsilon^4})$ respectively.

Given a query point $y$, let $b_j(y) := \{x_i : h_j(x_i) = h_j(y)\}$ be the set ("bin") of data points whose $h_j$-hash is the same as that of $y$. The estimator picks a uniformly random data point $x$ from $b_j(y)$ and computes $Z_j = \frac{1}{n}|b_j(y)| \cdot \sqrt{k(x, y)}$. If $b_j(y)$ is empty, then $Z_j = 0$. The final KDE estimate is $\widetilde{KDE}(y) = \frac{1}{L}\sum_{j=1}^{L} Z_j$. The query time is $O((T_H + T_k)/(\sqrt{\tau}\epsilon^2))$, where $T_k$ is the time it takes to evaluate $k$ on a single pair.

Our data structure is similar, except that for every $h_j$, we store the hash of every data point only with probability $\delta = 1/(n\sqrt{\tau})$. Therefore, on average we only compute and store a constant number of hashes of each data point, yielding expected preprocessing time of $O(T_H/(\tau \cdot \epsilon^2))$ and space usage of $O(S_H/(\tau \cdot \epsilon^2))$. The exact algorithm is given in Algorithm 1. Theorem 1, whose proof appears in the appendix, shows this still returns a sufficiently good estimate of $KDE_X(y)$.

**Example.** Let us give an illustration of the different approaches on the setting mentioned in the introduction. Suppose $\epsilon = \Theta(1)$ and $n \approx 1/\tau$. Consider a setting in which the query point is very close to a unique data point and very far from the rest of the data points. Concretely, $k(x_1, y) \approx 1$, while $k(x_i, y) \approx 0$ for every $i > 1$. The KDE value is $KDE_X(y) \approx \tau$. Naïve random sampling would have to sample $\Omega(1/\tau)$ points in order to pick up $x_1$ and return a correct estimate.

**Algorithm 1** : Space-Efficient HBE

---

**Proprocessing:**

---

**Input:** Dataset $X \subset \mathbb{R}^d$ of $n$ points; kernel $k(\cdot, \cdot)$; LSH family $H$; integer $1 \le L \le n$.
For $j = 1, \ldots, L$:
    Sample a random hash function $h_j$ from $H$.
    Let $X_j \subset X$ be a random subset that includes each point with independent probability $\frac{L}{n}$.[7]
    For every $x \in X_j$, evaluate and store $h_j(x)$.

---

**Query:**

---

**Input:** Query point $y \in \mathbb{R}^d$.
For $j = 1, \ldots, L$:
    Sample a uniformly random point $x^{(j)}$ from $b_j(y) = \{x \in X'_j : h_j(x) = h_j(y)\}$.
    Let $Z_j \leftarrow \frac{k(x^{(j)}, y) \cdot |b_j(y)|}{L \cdot \Pr_{h \sim H}[h(x^{(j)}) = h(y)]}$.
    Return $\frac{1}{L} \sum_{j=1}^{L} Z_j$.

---

In the HBE algorithm of [CS17], essentially in all hash tables $x_1$ would be the unique data point in the same bin as $y$, leading to a correct estimate $\frac{1}{L} \sum_{j=1}^{L} \frac{1}{n} |b_j(y)| \sqrt{k(x_1, y)} \approx \tau$. However, note that all terms in the sum are equal (to $\tau$), which seems to be somewhat wasteful. Indeed, it would suffice to pick up $x_1$ in just one hash table instead of all of them.

In our method, $x_1$ would be stored in $\delta L \approx 1$ hash tables in expectation, say only in $h_1$, and in that table it would be the unique data point in $b_1(y)$. In the other tables ($j > 1$) $b_j(y)$ would be empty, which means the estimator evaluates to zero. The resulting KDE estimate is $\frac{1}{L} \left( \frac{1}{n\delta} |b_j(y)| \sqrt{k(x_1, y)} + \sum_{j=2}^{L} 0 \right) \approx \tau$, which is still correct, while we have stored a hash of $x_1$ just once instead of $L$ times.

### 3.1 LSH for the Laplacian Kernel

The Laplacian kernel $k(x, y) = e^{-\|x-y\|_1 / \sigma}$ is a popular kernel, which fits naturally into the above framework since it is $(\frac{1}{2}, 1)$-LSHable. For simplicity, let us assume w.l.o.g. that in the dataset we need to hash, all point coordinates are in $[0, 1]$. This does not limit the generality since the Laplacian kernel is shift-invariant, and the coordinates can be scaled by inversely scaling $\sigma$.

The LSHablility of the Laplacian kernel follows from the Random Binning Features construction of Rahimi and Recht [RR07] (see details in the appendix). The expected hash size is $O(d \log(1/\sigma))$, and the hash evaluation time is $O(d)$. We also give a variant (described below) with better hash size and evaluation time for $\sigma \ge 1$. Together, the following lemma holds.

**Lemma 4.** *There is an LSH family $H_\sigma$ such that for every $x, y \in \mathbb{R}^d$, $\Pr_{h \sim H_\sigma}[h(x) = h(y)] = e^{-\|x-y\|_1/(2\sigma)}$. The expected hash size is $S_{H_\sigma} = O(\min\{d \log(1/\sigma), d/\sigma\})$ bits. The expected hashing time is $T_{H_\sigma} = O(\min\{d, d/\sigma\})$.*

The hashing family for the case $\sigma \ge 1$ is given as follows. Sample $\rho \sim \text{Poisson}(d/(2\sigma))$. Then sample $\xi_1, \ldots, \xi_\rho \in \{1, \ldots, d\}$ independently and uniformly at random, and $\zeta_1, \ldots, \zeta_\rho \in [0, 1]$ independently and uniformly at random. These random choices determine the hash function $h$. Next we describe $h$. Given a point $x$ to hash, for every $i = 1, \ldots, \rho$ set $b_i = 1$ if $x_{\xi_i} > \zeta_{\xi_i}$ and $b_i = 0$ otherwise. The hash $h(x)$ is the concatenation of $b_1, \ldots, b_\rho$. It is not hard to verify (see appendix) that $\Pr_h[h(x) = h(y)] = e^{-\|x-y\|_1/(2\sigma)}$.

Using the LSH family from Lemma 4 in Theorem 1 yields the following concrete data structure.

**Corollary 5** (Data structure for Laplacian KDE). *For the Laplacian kernel, there is a data structure for the KDE problem with expected space overhead $O(\min\{d \log(1/\sigma), d/\sigma\}/(\tau\epsilon^2))$, expected preprocessing time $O(\min\{d, d/\sigma\}/(\tau\epsilon^2))$, and query time $O(d/(\sqrt{\tau}\epsilon^2))$.*

Table 2: Properties of the datasets used in our experiments.

| Name | Description | Number of points | Dimension |
|------|-------------|------------------|-----------|
| Covertype | forest cover type | $581,012$ | 55 |
| Census | U.S. census | $2,458,285$ | 68 |
| GloVe | word embeddings | $1,183,514$ | 100 |
| MNIST | hand-written digits | $60,000$ | 784 |

## 4 Empirical Evaluation

We empirically evaluate our data structure for the Laplacian kernel.[8] For brevity, we will refer to the random sampling method as RS. The experimental results presented in this section are for the the Laplacian kernel $k(x,y) = e^{-\|x-y\|_1/\sigma}$. The results for the Exponential kernel are qualitatively similar are included in the appendix.

**Choice of datasets.** While the worst-case analysis shows that the HBE approach has asymptotically better query time than RS, it is important to note that RS can still attain superior performance in some practical settings. Indeed, the recent paper [SRB$^+$19] found this to be the case on various standard benchmark datasets, such as GloVe word embeddings [PSM14]. To reflect this in our experiments, we choose two datasets on which [SRB$^+$19] found HBE to be superior to RS as well as to state-of-the-art methods, and two datasets on which RS was found to be superior. The former two are Covertype [BD99] and Census[9], and the latter two are GloVe [PSM14] and MNIST [LC98]. Their properties are summarized in Table 2.

**Experimental setting.** We implement and evaluate Algorithm 1. Note that it is parameterized by the number of hash tables $L$, while its analysis in Theorem 1 is parameterized in terms of $\tau, \epsilon$, where we recall that $L = \Theta(1/(\sqrt{\tau}\epsilon^2))$. For practical implementation, parameterizing by $L$ is more natural since it acts as a smooth handle on the resources to accuracy tradeoff – larger $L$ yields better KDE estimates at the expense of using more time and space. $\tau, \epsilon$ need not be specified explicitly; instead, for any $\tau, \epsilon$ that satisfy $L = \Omega(1/(\sqrt{\tau}\epsilon^2))$, the guarantee of Theorem 1 holds (namely, for every query whose true KDE is at least $\tau$, the KDE estimate has up to $\epsilon$ relative error with high probability).

We compare our method to the HBE method of [CS17], as well as to RS as a baseline. The plots for HBE and our method are generated by varying the number of hash functions $L$. The plots for RS are generated by varying the sample size. Note that neither method has any additional parameters to set. For each method and each parameter setting, we report the median result of 3 trials. For each dataset we choose two bandwidth settings, one which yields median KDE values of order $10^{-2}$, and the other of order $10^{-3}$.[10] The bandwidth values and their precise method of choice are specified in the appendix. The appendix also includes accuracy results for varying bandwidth values (Fig. 9).

**Evaluation metrics.** We evaluate the query time, space usage and preprocessing time. In all of the plots, the y-axis measures the average relative error (which directly corresponds to $\epsilon$) of the KDE estimate, over 100 query points randomly chosen from the dataset. In the query time plots, the x-axis counts the number of kernel evaluations per query, which dominates and serves as a proxy for the running time. In the space usage plots, the x-axis counts the number of stored hashes. We use this measure for the space usage rather than actual size in bits, since there are various efficient ways to store each hash, and they apply equally to all algorithms. We also note that the plots do not account for the space needed to store the sampled dataset itself, which is the same for all methods. RS is not displayed on these plots since it has no additional space usage. In all three methods the preprocessing time is proportional to the additional space usage, and is qualitatively captured by the same plots.

**Results.** The query time plots consistently show that the query time to approximation quality tradeoff of our method is essentially the same as [CS17], on all datasets. At the same time, the space usage plots show that we have achieve a significantly smaller space overhead, with the gap from [CS17] substantially increasing as the target relative error becomes smaller. These findings affirm the direct advantage of our method as specified in Table 1.

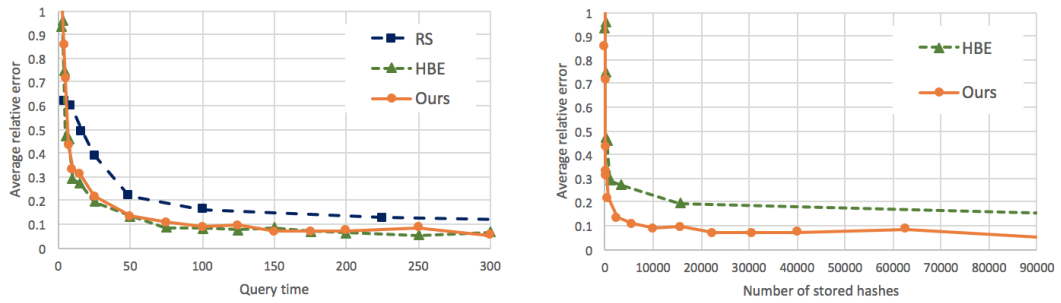

Figure 1: Covertype dataset, typical KDE values of order $10^{-2}$.

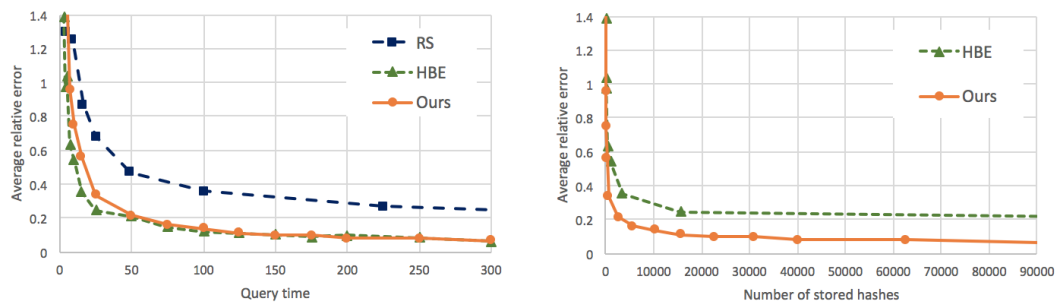

Figure 2: Covertype dataset, typical KDE values of order $10^{-3}$.

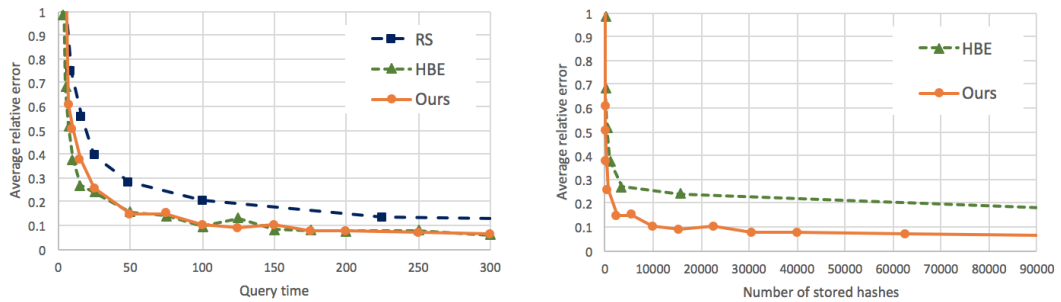

Figure 3: Census dataset, typical KDE values of order $10^{-2}$.

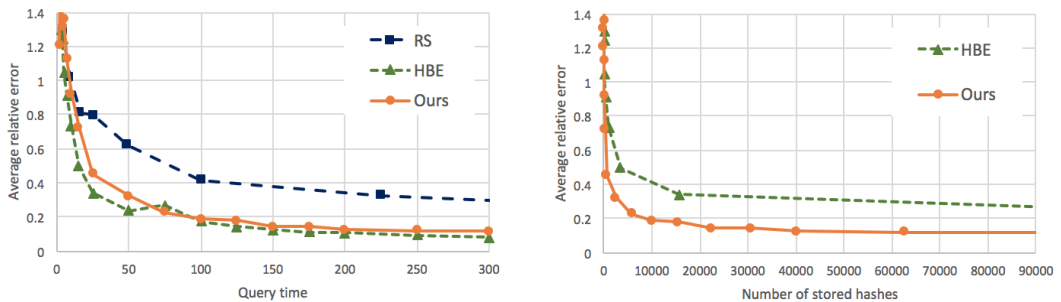

Figure 4: Census dataset, typical KDE values of order $10^{-3}$.

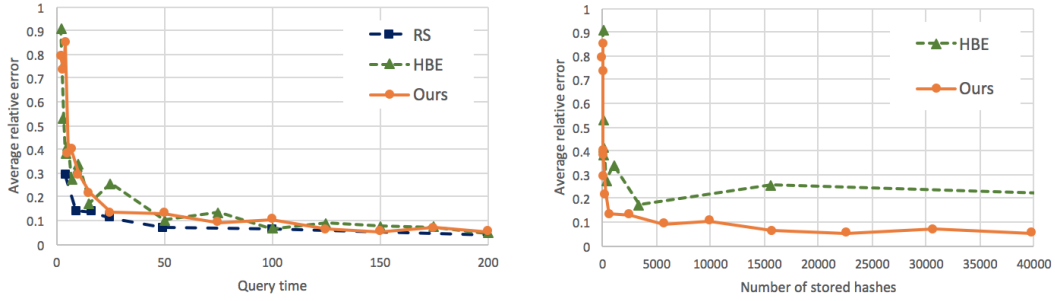

Figure 5: MNIST dataset, typical KDE values of order $10^{-2}$.

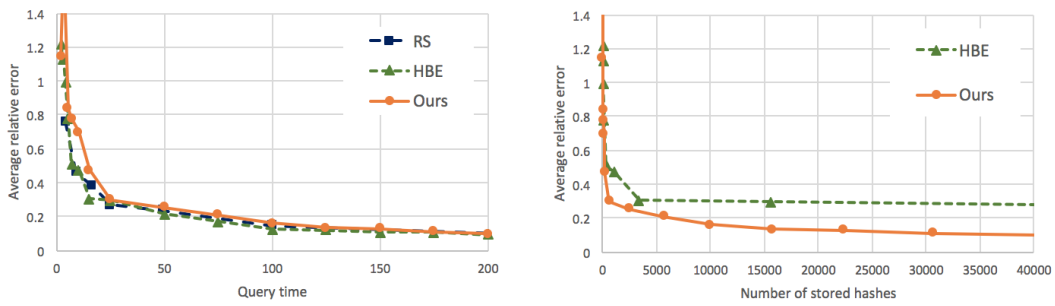

Figure 6: MNIST dataset, typical KDE values of order $10^{-3}$.

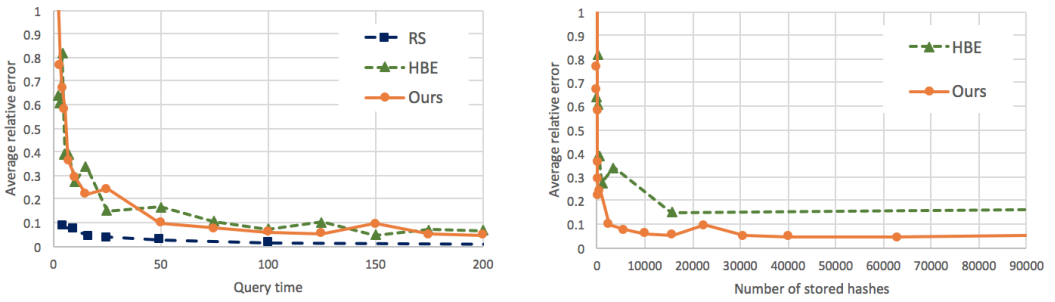

Figure 7: GloVe dataset, typical KDE values of order $10^{-2}$.

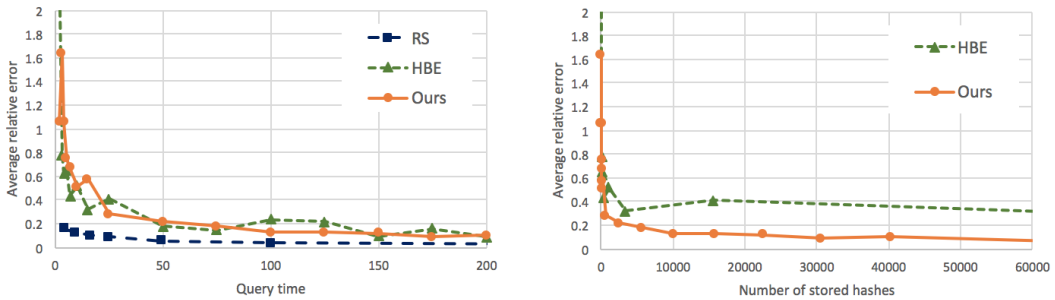

Figure 8: GloVe dataset, typical KDE values of order $10^{-3}$.

**Acknowledgments**

We thank the anonymous reviewers for useful suggestions. Piotr Indyk was supported by NSF TRIPODS award #1740751 and Simons Investigator Award.

## Footnotes

[4]For simplicity, the bounds in the table assume that the kernel takes $O(d)$ time to compute, and that a hash value takes $O(d)$ time to compute. The kernels we consider have these properties (for bandwidth $\sigma = \Omega(1)$). See Theorem 1 for the full parameter dependence.

[5]However, core-sets preserve all KDE values with high probability, while simple random sampling only preserves the KDE of any individual query with high probability.

[6]The HBE framework of [CS17] accommodates $(\beta, M)$-LSHable kernels, that satisfy $M^{-1} \cdot k(x, y)^\beta \leq \Pr_{h \sim H}[h(x) = h(y)] \leq M \cdot k(x, y)^\beta$, where $\beta$ can take any value in $[\frac{1}{2}, 1)$, and lower $\beta$ is better. Since the kernels we consider attain the optimal setting $\beta = \frac{1}{2}$, we fix this value throughout.

[7]This can be implemented in expected time $O(L)$ by sampling $\tilde{L} \sim \text{Binomial}(n, \frac{L}{n})$, and then sampling a uniformly random subset of size $\tilde{L}$.

[8]Our code is available at `https://github.com/talwagner/efficient_kde`.

[9]Available at `https://archive.ics.uci.edu/ml/datasets/US+Census+Data+(1990)`.

[10]In all the considered settings, the average KDE value is within a factor of at most 2 from the median KDE.

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
