[Supplementary Material · kde_space_supp.pdf]

# Space and Time Efficient Kernel Density Estimation in High Dimensions: Supplementary Material

**Arturs Backurs**[*]
TTIC
backurs@ttic.edu

**Piotr Indyk**
MIT
indyk@mit.edu

**Tal Wagner**
MIT
talw@mit.edu

---

[*]Authors ordered alphabetically.

# A  Proof of Theorem 1

Theorem 1 is a strengthening of the main results of [CS17]. Let us first describe their analysis.

Fix a dataset $X = \{x_1, \ldots, x_n\} \subset \mathbb{R}^d$, a query point $y \in \mathbb{R}^d$, and a kernel map $k : \mathbb{R}^d \times \mathbb{R}^d \to [0, 1]$. For every $i = 1, \ldots, n$, let $w_i = k(x_i, y)$.

Let $H$ be a family of hash functions from $\mathbb{R}^d$ to an arbitrary range $U$. For every $x_i$, denote its collision probability with $y$ by $p_i = \Pr_{h \sim H}[h(x_i) = h(y)]$. Let $b_h(y) = \{i : h(x_i) = h(y)\}$ be the set of points with the same hash as $y$. Suppose we have $M \geq 1$ and $\beta \in [\frac{1}{2}, 1)$ such that for every $i$, $M^{-1} \cdot w_i^\beta \leq p_i \leq M \cdot w_i^\beta$. (It is instructive to think of the case $M = 1$ and $\beta = \frac{1}{2}$.)

The KDE estimator of [CS17] is $Z = \frac{w_i \cdot |b_h(y)|}{n \cdot p_i}$, where $i$ is chosen uniformly at random from $b_h(y)$. If $b_h(y)$ is empty, then $Z = 0$.

**Theorem 6** ([CS17]). $\mathbb{E}[Z] = \mathrm{KDE}_X(y)$, and $\mathrm{Var}[Z] \leq M^3 \cdot \mathrm{KDE}_X(y)^{2-\beta}$.

Since the dependence of $\mathrm{Var}[Z]$ on $\mathbb{E}[Z]$ is strictly better than quadratic, one can use this estimator to estimate $\mathrm{KDE}_X(y)$ with a smaller number of samples than naïve random sampling. In particular, if $\tau > 0$ is a lower bound $\mathrm{KDE}_X(y)$, then in order to get a multiplicative $(1 \pm \epsilon)$-approximation, the sufficient number of samples is $\mathrm{Var}[Z]/(\epsilon \, \mathbb{E}[Z])^2 = O(M^3/(\tau^\beta \epsilon^2))$. If the kernel admits an LSH family with good parameters $\beta$ and $M$, then this is better than random sampling, which would require $O(1/(\tau \epsilon^2))$ samples. This is the driving force behind the HBE method of [CS17].

To obtain Theorem 1, we hash each point only with probability $\delta = 1/(n\tau^{1-\beta})$, where $\tau \leq \mathrm{KDE}_X(y)$. Formally, let $r_1, \ldots, r_n$ be i.i.d. Bernoulli random variables with $\Pr[r_i] = \delta$. Let

$$b_h'(y) = \{i : h(x_i) = h(y) \text{ and } r_i = 1\}$$

be sparsified counterpart of $b_h(y)$. Our modified KDE estimator is $Z' = \frac{w_i \cdot |b_h'(y)|}{n \cdot \delta \cdot p_i}$, where $i$ is chosen uniformly at random from $b_h'(y)$. If $b_h'(y)$ is empty, then $Z' = 0$. We prove the following.

**Theorem 7.** $\mathbb{E}[Z'] = \mathrm{KDE}_X(y)$, and $\mathrm{Var}[Z'] \leq (M^3 + M) \cdot \mathrm{KDE}_X(y)^{2-\beta}$.

*Proof.* Our proof closely follows that of Theorem 6. Starting with the expectation,

$$
\begin{aligned}
\mathbb{E}[Z'] &= \frac{1}{n\delta} \mathbb{E} \, \frac{w_i}{p_i / |b_h'(y)|} \\
&= \frac{1}{n\delta} \mathop{\mathbb{E}}_{\substack{r_1, \ldots, r_n \\ h \sim H}} \mathop{\mathbb{E}}_{i \in b_h'(y)} \left[ |b_h'(y)| \frac{w_i}{p_i} \right] \\
&= \frac{1}{n\delta} \mathop{\mathbb{E}}_{\substack{r_1, \ldots, r_n \\ h \sim H}} \sum_{i \in b_h'(y)} \frac{w_i}{p_i} \\
&= \frac{1}{n\delta} \sum_{i=1}^n \frac{w_i}{p_i} \mathop{\Pr}_{\substack{r_1, \ldots, r_n \\ h \sim H}} [i \in b_h'(y)] \\
&= \frac{1}{n\delta} \sum_{i=1}^n \frac{w_i}{p_i} \mathop{\Pr}_{\substack{r_1, \ldots, r_n \\ h \sim H}} [i \in b_h(y) \ \& \ r_i = 1] \\
&= \frac{1}{n} \sum_{i=1}^n w_i \\
&= \mathrm{KDE}_X(y).
\end{aligned}
$$

Next we bound the variance:

$$\mathrm{Var}[Z'] \leq \mathbb{E}[(Z')^2]$$

$$= \frac{1}{n^2\delta^2} \mathbb{E}\left[\frac{w_i^2}{p_i^2/|b_h'(y)|^2}\right]$$

$$= \frac{1}{n^2\delta^2} \mathop{\mathbb{E}}_{\substack{r_1,\dots,r_n \\ h\sim H}} \mathop{\mathbb{E}}_{i\in b_h'(y)}\left[|b_h'(y)|^2 \frac{w_i^2}{p_i^2}\right]$$

$$= \frac{1}{n^2\delta^2} \mathop{\mathbb{E}}_{\substack{r_1,\dots,r_n \\ h\sim H}}\left[|b_h'(y)| \sum_{i\in b_h'(y)} \frac{w_i^2}{p_i^2}\right]$$

$$= \frac{1}{n^2\delta^2} \mathop{\mathbb{E}}_{\substack{r_1,\dots,r_n \\ h\sim H}}\left[\sum_j [j\in b_h'(y)] \sum_i [i\in b_h'(y)] \frac{w_i^2}{p_i^2}\right]$$

$$= \frac{1}{n^2\delta^2} \sum_i \frac{w_i^2}{p_i^2} \sum_j \mathop{\mathbb{E}}_{\substack{r_1,\dots,r_n \\ h\sim H}}[[j\in b_h'(y)][i\in b_h'(y)]]$$

$$= \frac{1}{n^2\delta^2} \sum_i \frac{w_i^2}{p_i^2} \sum_j \mathop{\mathrm{Pr}}_{\substack{r_1,\dots,r_n \\ h\sim H}}[j\in b_h'(y) \ \& \ i\in b_h'(y)].$$

We split the last term into two expressions:

$$\frac{1}{n^2\delta^2} \sum_i \frac{w_i^2}{p_i^2} \sum_{j:j\neq i} \mathop{\mathrm{Pr}}_{\substack{r_1,\dots,r_n \\ h\sim H}}[j\in b_h'(y) \ \& \ i\in b_h'(y)], \tag{1}$$

and

$$\frac{1}{n^2\delta^2} \sum_i \frac{w_i^2}{p_i^2} \mathop{\mathbb{E}}_{\substack{r_1,\dots,r_n \\ h\sim H}}[i\in b_h'(y)]. \tag{2}$$

To upper bound Eq. (2), we observe that, since $j\neq i$,

$$\mathop{\mathrm{Pr}}_{\substack{r_1,\dots,r_n \\ h\sim H}}[j\in b_h'(y) \ \& \ i\in b_h'(y)] = \delta^2 \mathop{\mathrm{Pr}}_{h\sim H}[j\in b_h(y) \ \& \ i\in b_h(y)] \leq \delta^2 p_j.$$

Therefore, Eq. (2) is upper bounded by $\frac{1}{n^2}\sum_i \frac{w_i^2}{p_i^2}\sum_j p_j$. This expression is bounded in the proof of Theorem 6 in [CS17], and we now reproduce the argument for completeness. We observe that $\Pr[j\in b_h(y) \ \& \ i\in b_h(y)] \leq \Pr[j\in b_h(y)] = p_j$ and, using the bounds $\frac{w_i^\beta}{M} \leq p_i \leq M w_i^\beta$, conclude that

$$\frac{1}{n^2} \sum_i \frac{w_i^2}{p_i^2} \sum_j p_j \leq \frac{M^3}{n^2} \sum_i w_i^{2-2\beta} \sum_j w_j^\beta.$$

To prove $\mathrm{Var}[Z] \leq M^3 \mathrm{KDE}_X(y)^{2-\beta}$, it is sufficient to show

$$\frac{1}{n^2} \sum_i w_i^{2-2\beta} \sum_j w_j^\beta \leq \left(\frac{1}{n}\sum_i w_i\right)^{2-\beta}.$$

This follows from the inequalities $\frac{1}{n}\sum_i w_i^{2-2\beta} \leq \left(\frac{1}{n}\sum_i w_i\right)^{2-2\beta}$ and $\frac{1}{n}\sum_j w_j^\beta \leq \left(\frac{1}{n}\sum_i w_i\right)^\beta$. The first inequality holds for any $\beta$ that satisfies $0\leq 2-2\beta\leq 1$ and the second inequality holds for any $0\leq\beta\leq 1$. That is, both inequalities hold if $\frac{1}{2}\leq\beta\leq 1$.

To upper bound Eq. (3) we observe that $\mathbb{E}_{\substack{r_1,\dots,r_n \\ h\sim H}}[i\in b_h'(y)] = p_i\delta$ and therefore Eq. (3) is upper bounded by $\frac{1}{n^2\delta}\sum_i \frac{w_i^2}{p_i} \leq \frac{M}{n^2\delta}\sum_i w_i^{2-\beta}$. Since $\sum_i w_i^{2-\beta} \leq \sum_i w_i = n\cdot\mathrm{KDE}_X(y)$ and $\delta = 1/(n\tau^{1-\beta}) \geq 1/(n\cdot\mathrm{KDE}_X(y)^{1-\beta})$, this is upper bounded by $M\cdot\mathrm{KDE}_X(y)^{2-\beta}$, as needed. $\qquad\square$

To derive Theorem 1, set $\beta = 1/2$. By the above theorem, the estimator $Z'$ is unbiased and satisfies $\mathrm{Var}[Z'] = 2M^3/\tau^{1.5}$. Therefore, in order to obtain a $(1\pm\epsilon)$-approximation for $\mathrm{KDE}_X(y)$, it is sufficient to return the average over $L = O(M^3/(\sqrt{\tau}\epsilon^2))$ independent samples of the estimator.

**Preprocessing time:** To be able to draw samples from the estimator, we need to hash a subset of the $n$ pointset $x_1, \ldots, x_n$. The expected size of the subset is $\delta n = 1/\sqrt{\tau}$. The time needed to hash a single point is $T_H$. We need to repeat this $L$ times (once for each sample of the estimator we would draw int he query phase). The total preprocessing time complexity becomes

$$1/\sqrt{\tau} \cdot T_H \cdot L = O\left(\frac{1}{\tau} \cdot \frac{T_H M^3}{\varepsilon^2}\right).$$

**Space usage:** In order to draw a single sample from the estimator, we store the hash of each point $x_i$ for which $r_i = 1$. We also need to fully store $x_i$ itself, since if we draw it from $b'_h(y)$ during the query phase, we would need to evaluate $k(x_i, y)$ in order to compute $Z'$. The expected numbers of these points is $\delta n = 1/\sqrt{\tau}$, so for a single sample we store in expectation $(S_X + S_H)/\sqrt{\tau}$ bits, where $S_X$ is the storage size of a data point, and $S_H$ is the storage size of a hash value. Repeating this $L$ times, the total storage size is

$$\frac{S_X + S_H}{\sqrt{\tau}} \cdot L = O\left(\frac{1}{\tau} \cdot \frac{(S_X + S_H)}{\varepsilon^2}\right) \text{ bits.}$$

**Query time:** To draw a sample from the estimator, we need to hash the query point $y$. This takes time $T_H$. Furthermore, given the hash value, we need to sample a random element from the corresponding bucket $b'_h(y)$ and evaluate the random variable $Z'$. This takes time $T_k$. Thus, we spend $T_H + T_k$ time to draw a single sample from the estimator. Since we do that $L$ times, the total query time is

$$L \cdot (T_H + T_k) = O\left(\frac{1}{\sqrt{\tau}} \cdot \frac{(T_H + T_k)M^3}{\varepsilon^2}\right)$$

as promised.

# B  KDE Data Structure for the Gaussian Kernel

For the Gaussian kernel $k(x, y) = e^{-\|x-y\|^2/\sigma^2}$, the best LSHability result we are aware of is based on the ball-carving LSH of [AI06].

**Theorem 8** ([AI06]; see also Theorem 11 in [CS17]). *For any $R > 0$ there exists a distribution $H$ of hash functions such that for any $x, y \in \mathbb{R}^d$ with $\|x - y\|_2 \leq R$ the following bounds hold.*

$$e^{-\|x-y\|_2^2} \cdot e^{-O(R^{4/3} \log R)} \leq \Pr_{h \sim H}[h(x) = h(y)] \leq e^{-\|x-y\|_2^2} \cdot e^{O(R^{4/3} \log R)}.$$

*The time complexity of computing a hash value $h(x)$ is $d \cdot e^{O(R^{4/3} \log R)}$. Finally, the probability $Pr_{h \sim H}[h(x) = h(y)]$ is non-increasing in the distance $\|x - y\|_2$.*

It can be used to give the following time and space efficient data structure for Gaussian KDE.

**Theorem 9.** *Given $n$ points $y_1, \ldots, y_n \in \mathbb{R}^d$ and parameters $1 \geq \tau \geq \frac{1}{n^2}$ and $1 \geq \varepsilon \geq \frac{1}{n^2}$, we can build a data structure in space $\frac{1}{\tau} \cdot \frac{n^{o(1)}}{\varepsilon^4}$ that efficiently answers $\text{KDE}(x)$ queries for the Gaussian kernel $k(x, y) = e^{-\|x-y\|_2^2}$. In particular, given a query point $x \in \mathbb{R}^d$ with $\text{KDE}(x) \geq \tau$, we can approximate $\text{KDE}(x)$ within the multiplicative factor of $1 + \varepsilon$ in time $O(d) \cdot \frac{\log^3 n}{\varepsilon^2} + \frac{1}{\sqrt{\tau}} \cdot \frac{n^{o(1)}}{\varepsilon^4}$.*

*Proof.* The proof proceeds in two steps. First apply Theorem 1 to the above LSHability result. We use Theorem 8 and set $R = (\log n)^{2/3}$. We get that the hashing scheme $H$ satisfies

$$e^{-\|x-y\|_2^2} \cdot n^{-o(1)} \leq \Pr_{h \sim H}[h(x) = h(y)] \leq e^{-\|x-y\|_2^2} \cdot n^{o(1)}$$

for all $x, y \in \mathbb{R}^d$ with $\|x - y\|_2 \leq R = (\log n)^{2/3}$. Furhermore, the hashing can be performed in $d \cdot n^{o(1)}$ time.

We can get rid of the assumption that $\|x - y_i\|_2 \leq R$ for all $i = 1, \ldots, n$ as follows. Observe that, if $\|x - y_i\|_2 > R$, then $\Pr_{h \sim H}[h(x) = h(y_i)] \leq e^{-R^2} n^{o(1)} \leq e^{-(\log n)^{4/3}} n^{o(1)} \leq n^{-\omega(1)}$. We used the fact that the probability $\Pr_{h \sim H}[h(x) = h(y_i)]$ is non-increasing in the distance $\|x - y_i\|_2$. This

implies that with probability $1 - n^{\omega(1)}$ we have that all $i$ with $h(x) = h(y_i)$ satisfy $\|x - y_i\|_2 \leq R$. Since all our samples $y_i$ satisfy $h(x) = h(y_i)$, we lose at most a neglibile factor in the probability of success.

We can get a KDE algorithm by setting $M = n^{o(1)}$, $T_H = dn^{o(1)}$, $S_X = S_H = O(d \log n)$ and $T_k = O(d)$.

In the second step, we improve the dependence on $d$ by dimension reduction. In particular, we reduce the space by projecting the points from the $d$ dimensional space to $\frac{O(\log^3 n)}{\varepsilon^2}$ dimensional space. The extra term $O(d) \cdot \frac{\log^3 n}{\varepsilon^2}$ in the time complexity comes from the time needed to perform the projection.

We randomly project the points $y_i$ and the point $x$ to $O(\log n)/\delta^2$ dimensional space for $\delta = \frac{\varepsilon}{2 \ln(1/(\varepsilon\tau))}$. This preserved all distances $\|x - y_i\|$ within the multiplicative factor of $1 \pm \delta$. After the projection, the contribution from a point $y_i$ to the KDE value becomes $\exp(-(1 \pm \delta)\|x - y_i\|_2^2)$. Consider the case $\|x - y_i\|_2^2 \geq 2 \ln(1/(\varepsilon\tau))$. The contribution of such a point $y_i$ after the projection is $\leq \exp(-\|x - y_i\|_2^2/2) \leq \varepsilon\tau$. Thus, the average contribution from such point to the KDE value after the projection is $\varepsilon\tau$, which can be subsumed by the $1 + \varepsilon$ multiplicative approximation. Consider the case $\|x - y_i\|_2^2 < 2 \ln(1/(\varepsilon\tau))$. In this case we observe that after the projection the contribution $\exp(-(1 \pm \delta)\|x - y_i\|_2^2) = \exp(-\|q - p_i\|_2^2) \exp(\pm\delta\|x - y_i\|_2^2)$ differs from the original contribution by a multiplicative factor of at most $\exp(\delta\|x - y_i\|_2^2) \leq 1 + O(\varepsilon)$ since $\delta\|x - y_i\|_2^2 \leq \varepsilon \leq 1$. Therefore, in this case too we introduce a multiplicative error of at most $1 + \varepsilon$.

This allows us to reduce the dimensionality of the pointset from $d$ to
$$\frac{O(\log n)}{\delta^2} = \frac{O(\log n) \log(1/(\varepsilon\tau))^2}{\varepsilon^2} \leq \frac{O(\log^3 n)}{\varepsilon^2}$$
for the purpose of estimating $\mathrm{KDE}_X(y)$. $\qquad\qquad\square$

## C   Laplacian Kernel LSH

In this section we fully describe the LSHability of the Laplacian kernel, as per Lemma 4. Recall that we assume w.l.o.g. that all point coordinates are in $[0, 1]$. For the sake of clarity, we will describe LSH families $H$ such that $\Pr_{h \sim H}[h(x) = h(y)] = e^{-\|x-y\|_1/\sigma}$. The $(\frac{1}{2}, 1)$-LSHable property then follows simply by doubling the bandwidth $\sigma$.

For $\sigma < 1$ we use the Random Binning Features of Rahimi and Recht [RR07], which we now recall. Start with the one-dimensional case $d = 1$. Sample $c$ from the Gamma distribution with shape 2 and scale $\sigma$. The probability density function of this distribution is $p(x) = \sigma^{-2} \cdot x \cdot e^{-x/\sigma}$. Then, impose on the real line a one-dimensional uniform grid of side length $c$, shifted by a uniformly random $s \sim [0, c)$. The random choices of $c$ and $s$ determine the hash function $h$. Given a point $x \in [0, 1]$, $h$ maps it to the grid cell containing it.

One can verify that $\Pr_h[h(x) = h(y)] = e^{-|x-y|/\sigma}$ for every $x, y \in [0, 1]$ [RR07], and that the time to evaluate $h(x)$ is $O(1)$. Furthermore, the number of grid cells intersecting the interval $[0, 1]$ is $\Theta(1/c)$. Since $1/c$ has an inverse-Gamma distribution, its expected value is $1/\sigma$, hence there are $\Theta(1/\sigma)$ grid cells in expectation, and thus the expected space to store a hash value is $\log(1/\sigma) + O(1)$ bits. Finally, for an arbitrary dimension $d$, we simply perform the above for each dimension independently, and concatenate the resulting hashes. We then have $\Pr_h[h(x) = h(y)] = e^{-\|x-y\|_1/\sigma}$ with hash evaluation time $O(d)$ and expected hash size $O(d \log(1/\sigma))$.

For $\sigma \geq 1$, we use the LSH family described in Section 3.1. Start with the one-dimensional case $d = 1$. For a uniformly random $\zeta \in [0, 1]$, let $b(x) = 1$ if $x > \zeta$ and $b(x) = 0$ otherwise, and similarly $b(y) = 1$ if $y > \zeta$ and $b(y) = 0$ otherwise. Then we have $\Pr[b(x) = b(y)] = 1 - |x - y|$. In the arbitrary dimensional case $x, y \in [0, 1]^d$, applying this to a uniformly random dimension $\xi \in \{1, \ldots, d\}$ yields $\Pr[b(x) = b(y)] = \frac{1}{d} \sum_{\xi=1}^{d} (1 - |x_\xi - y_\xi|) = 1 - \frac{1}{d}\|x - y\|_1$. If we repeat this $\rho$ independent times, where $\rho$ is a fixed non-negative integer, and let $h(x)$ be the concatenation of the $b(x)$'s of the $\rho$ repetitions (and similarly define $b(y)$), then $\Pr[h(x) = h(y)] = (1 - \frac{1}{d}\|x - y\|_1)^\rho$. Finally, choosing $\rho \sim \mathrm{Poisson}(d/\sigma)$ yields
$$\Pr[h(x) = h(y)] = \sum_{\rho=0}^{\infty} \frac{e^{-d/\sigma} \cdot (d/\sigma)^\rho}{\rho!} \cdot (1 - \frac{1}{d}\|x - y\|_1)^\rho = e^{-\|x-y\|_1/\sigma}.$$

Table 1: Bandwidth settings used in our experiments.

| Dataset | Estimate of median NN distance ($\phi$) | Bandwidth for KDE values $\sim 10^{-2}$ | Bandwidth for KDE values $\sim 10^{-3}$ |
|---------|------------------------------------------|------------------------------------------|------------------------------------------|
| Covertype | 0.005 | $20 \cdot \phi$ | $10 \cdot \phi$ |
| Census | 0.01 | $5 \cdot \phi$ | $3 \cdot \phi$ |
| GloVe | 4.48 | $0.5 \cdot \phi$ | $0.25 \cdot \phi$ |
| MNIST | 38.1 | $1 \cdot \phi$ | $0.5 \cdot \phi$ |

## D    Additional Exprimental Details

**Bandwidth selection.**    The rule of thumb suggested in [JDH99] for bandwidth selection is to take the median distance of a query point to its nearest neighbor in the dataset. We estimate this parameter for each dataset and denote it by $\phi$. Since its effect of the KDE values is inconsistent between the various datasets, we scale it by a constant so as to make the typical KDE values be within a certain order of magnitude. Specifically, we experiment with two orders of magnitude, $10^{-2}$ and $10^{-3}$. (Note that larger typical values are essentially trivial to estimate by standard concentration inequalities, while for smaller values an approximation is largely uninformative). The specific numbers used are listed in Table 3. Note that the listed values of $\phi$ are estimated after shifting and scaling each dataset such that all point coordinates in are in $[0, 1]$.

**Accuracy with varying bandwidth.**    Figure 9 displays the accuracy on the Covertype and Census datasets for varying bandwidth values. The results are with $L = 250$ (i.e., each KDE value is estimated using 250 kernel evaluations). It shows that the accuracy of our method is similar to HBE, and significantly better then RS (whose accuracy improves and converges to the hashing-based methods as the the bandwidth grows and the KDE values become bounded away from 0). At the same time, the space usage of our method is smaller than HBE by a factor of $L = 250$.

Figure 1: Accuracy with $L = 250$ and varying bandwidth, for the Covertype (left) and Census (right) datasets. The space usage (not displayed in the plot) of HBE is larger by a factor of 250 than ours.

**Exponential kernel.**    Section 4 presented empirical results for the Laplacian kernel, and mentioned that similar results are achieved for the Exponential kernel. Some of these results are depicted in the figures below. The results for both hashing-based methods (HBE and ours) are obtained by a random rotation of the dataset[2] followed by the algorithm presented in the main text. Ground-truth KDE and RS are computed directly on the original $\ell_2$-distances.

Figure 2: Covertype dataset, Exponential kernel, typical KDE values of order $10^{-2}$.

Figure 3: Covertype dataset, Exponential kernel, typical KDE values of order $10^{-3}$.

Figure 4: Census dataset, Exponential kernel, typical KDE values of order $10^{-3}$.

Figure 5: Census dataset, Exponential kernel, typical KDE values of order $10^{-4}$.

## Footnotes

[2]It is known that the $\ell_2$-distances after a random rotation are approximately equal, with high probability, to the $\ell_1$-distances before the projection.