[Reviews · NeurIPS 2019]

Reviewer 1



This is overall an interesting direction but the impact of the proposed scheme is somewhat underdeveloped given that the gains are somewhat limited and most of the theoretical results build directly off of existing work. - While the proposed scheme does improve the storage requirements of HBE, it is not clear from empirical evaluation as to how much savings the proposed scheme provides relative to the data set size. The data set size and the O(1/\epsilon^2) factor appears to be the dominating storage requirement here (either the full or the subsampled set); it is not clear from the empirical results how significant are the O(1 / \sqrt{\tau}) savings here. - The example for the proposed scheme on page 5 (lines 150-153), the resulting KDE estimate is only accurate if L = 1/\sqrt{\tau} instead of being O(1 / \sqrt{\tau}), which requires use to know \tau beforehand which might be problematic. The intuition behind this example is very useful but maybe this explanation can be further clarified to address the aforementioned discrepancy. - The theoretical and empirical results demonstrate the ability of the proposed scheme to match the average performance of the existing HBE scheme. However, as mentioned in the manuscript, the variance of the estimates from the proposed scheme would be adversely affected. But this loss in variance is not empirically studied. ===== Post response ----- I thank the authors for the response. Based on the response and the discussion, I am motivated to raise the score.

Reviewer 2



Originality: This is a clever variant of the original HBE by Charikar and Siminelakis. Though it is simple and natural in hindsight, the fact is that nobody realised this in the last two years even though [CS17] appeared in a top venue. Quality: Proofs (in supplementary material) appear solid. The paper is generally very well written, but comparison of space usage for different methods is sloppy, making the improvement over [CS17] seem larger than it is. Many bounds stated seem to assume d=O(1). Table 1 is misleading. The issue is that it is only necessary to store each point once, and then use pointers to refer to it. Thus, the space usage of (a trivial modification of) the [CS17] HBE is d/τ · 1/ε^2 + 1/τ^{3/2} · 1/ε^2. The new bound is only better when d < τ^{1/2}. This should be made clear. Clarity: The paper is clearly written. I think it would be possible to reproduce all results. Significance: I believe this is a significant improvement over [CS17] as well as over the variant of in the recent ICML paper [SRB+19]. Since the problem is of wide importance, I think this will be of interest to many NeurIPS participants.

Reviewer 3



This paper is relevant since hashing is important for high-dimensional data, and the proposed method is a simple variant of HBE. The idea of introducing sparseness in estimation is a popular technique in many problems, so the methodology itself is not new. But, adopting it to this setting is a nice attempt, and the space saving should be valuable in practice. The theoretical part is not that strong, perhaps because the idea is simple. Two questions: 1) why is the sampling probability chosen as \delta=1/(n\sqrt\tau)? I didn’t find a clear explanation in the paper. 2) In the theorems and lemmas, by ‘’there exists a data structure…’’ you just refer to the proposed one, right? Post Rebuttal: After reading the rebuttal and re-reading the paper, I am willing to raise the score from 5 to 6.

Reviewer 4



Overall the paper is an average paper but clearly written. This paper proposes an improvement of Charikar's approach to achieve sublinear kernel density estimation with linear space and linear time preprocessing. Experimental results focus mainly on Laplacian (L1 variant in the main submission and L2 variant added in supplement). The key observation for achieving linear space is to modify the previous HBE approach so that each hash table stores each point in the dataset with constant probability - in this way, the superlinear storage cost is overcome. However, my main complaint is in the experimental results. First of all, the authors need to clarify what is meant by "typical kernel value" - I understand the median nearest neighbor distance is estimated and then subsequently scaled to obtain the bandwidth value used for the kernel. By "typical", does it mean 50 % of the dataset (or any other threshold)? What about the distribution of the raw kernel values themselves and the skewness/kurtosis? From the plots, it is impossible to judge how these factors are accounted to achieve scalability. I think it would be better for the authors to present a more complete picture by showing the scalability as the bandwidth of the kernel is varied. In summary: Originality: builds upon the previous hashing-based approach by Charikar et al., though the modification is trivial. Quality: average - I would like to have seen more experimental results. Clarity: The paper is clearly written - though it may be good idea for the authors to correct minor punctuation errors and run-on sentences. Significance: space improvement over the previous approach is novel but the experimental results need to be supplemented for demonstrating a stronger message.

[Author Response · NeurIPS 2019]

We thank the reviewers for their useful feedback and suggestions.

**Significance of space improvement:** The need to reduce the space requirements and the preprocessing time of the
[CS17] algorithm was pointed out previously in the literature. For example [SRB+] states that *"...straightforward*
*implementations of the proposed approach require a large amount [...] of preprocessing time and memory to create the*
*requisite number of hash tables"* (Limitations section, pg. 2). This motivated [SRB+] to develop core-set-like methods
for reducing the data size (cf. lines 93–99 in our submission).

Regarding theoretical analysis, we emphasize that our result is a direct theoretical improvement to the framework of
[CS17]. While the algorithmic modification is simple (which has the advantage of being practical), the analysis is
rather subtle (cf. Theorem 7). Moreover, it is nearly optimal: As mentioned on page 3, any algorithm must use at least
$N = \Omega(1/(\tau \cdot \epsilon^2))$ data samples to guarantee $(1 + \epsilon)$-approximation (see e.g., [PT18]). Thus, the $1/\sqrt{\tau}$ improvement
that we provide is essentially the best possible, as our storage is proportional to the size $N$ of the sample data set. We
remark, however, that further improvements could be still possible by storing $O(N)$ bits instead of memory words.

Regarding the quantitative improvement, we note that the lower bound $\tau$ can be very close to zero in practice, which
makes the $1/\sqrt{\tau}$ improvement very substantial. Concretely, if in Algorithm 1 we set the budget of kernel computations
per KDE query to $L$ (recall that analytically $L \sim 1/\sqrt{\tau}$), then our method stores $L$ hashes per point (in expectation)
while HBE stores $L^2$. Since $L$ is typically of the order of at least dozens or hundreds, the gap in performance can be
large. For example, the plots attached to this file are for $L = 250$, resulting in space reduction by a factor of 250.

**Additional experiments:** We thank you for the useful
suggestion to study the performance as the bandwidth is
varied. We have run those experiments and representa-
tive results are included here (as space permits). We will
include the full results in the revised version. The plots
shown here are for 250 kernel evaluations per query with
varying bandwidths. The x-axis values in boldface are the
bandwidth values used in our submission. As predicted,
(a) HBE and our method achieve similar error, (b) our
space usage is significantly better (by a factor of 250, as
explained above), (c) as the bandwidth grows and the
KDE values become bounded away from 0 the estimation
task becomes easier, and in particular naïve random sam-
pling improves and converges to the same error as HBE.
We will also include results for other LSHable kernels.

**Units of space:** The plots measure space in single bits (however, the labels on the x-axis are accidentally scaled by 100;
we have fixed this). We accept the reviewer's remark that there are various efficient ways to store each hash, so we redid
the experiments by measuring the *number* of stored hashed points. The resulting plots are essentially scaled versions of
the plots in the paper. In particular, the relative improvement (by a factor of $L$) over HBE is the same. For the sake of
compatibility with our original submission, the plots in this rebuttal still measure space in bits.

**Clarifications:** All of the following points raised by the reviewers will be revised or clarified in the final version.
Apologies for any confusion. **R1:** "scheme on page 5" Correct. More accurately for any $L$, the expected number of
non-empty bins $b_j(y)$ is $\delta L$, and the estimate in line 152 has expectation $\tau$, without having to know it. **R1:** "loss in
variance" The variance affects the error of the estimate, so its impact is indirectly reported on Figs 1–8. We believe the
*relative* error reported on those figures is more natural in our context than the additive squared error (i.e., the empirical
variance), because it directly corresponds to the parameter $\epsilon$. **R2:** "stated bound seem to assume $d = O(1)$" Indeed,
the introduction alternates between the number of kernel evaluations and the running time, not always consistently.
Generally, the running time (either of the preprocessing or the kernel evaluation) should have a factor of $d$ in front of it.
**R2:** "The new bound is only better when $d < \tau^{1/2}$" Indeed, the second column of Table 1 accurately describes the
*preprocessing time* of both algorithms, as opposed to their extra space usage (which can be reduced for both algorithms
by storing pointers not points). Theorem 1 summarizes the bounds correctly, parameterized by the relevant quantities.
**R2:** "additive $\epsilon\tau$ error guarantee?" This is a subtle point. Our understanding is that without additional assumptions,
such additive error guarantee must require $1/(\epsilon\tau)^2$ samples by anti-concentration. Whether it can be achieved under
weaker assumptions needs further exploration. **R2:** "strictly better than quadratic?" Correct. **R2:** "query points from
the data set?" Yes. **R3:** "why is the sampling probability chosen as $\delta = 1/(n\sqrt{\tau})$?" Intuitively, so each element appears
in expectation in $O(1)$ hash tables (cf. lines 71–75). Technically, this equates the variance term of our modification to
that of the original HBE algorithm, ensuring that our variance is at most twice that of HBE (cf. last two lines in proof of
Theorem 7, with $\beta = 0.5$). **R3:** "refer to the proposed one, right?" Correct. **R4:** "what is meant by "typical kernel
value"" Median KDE (which for all the considered settings is within a factor of 2 from the average KDE).

[Meta-Review · NeurIPS 2019]

After a careful discussion among the reviewers, there is a clear consensus that the paper provides a solid theoretical contributions for kernel density estimation in high dimensions. Hence, I am happy to recommend to accept this paper for publication at NeurIPS2019. Nevertheless, one concern that came up during the discussion is the lack of experimental comparison and clarity in how it was performed. I urge the authors to incorporate reviewers' comment on this weakness of the paper into the camera-ready version.